# Flavonoids as Potential Wound-Healing Molecules: Emphasis on Pathways Perspective

**DOI:** 10.3390/ijms24054607

**Published:** 2023-02-27

**Authors:** Nabilah Zulkefli, Che Nur Mazadillina Che Zahari, Nor Hafiza Sayuti, Ammar Akram Kamarudin, Norazalina Saad, Hamizah Shahirah Hamezah, Hamidun Bunawan, Syarul Nataqain Baharum, Ahmed Mediani, Qamar Uddin Ahmed, Ahmad Fahmi Harun Ismail, Murni Nazira Sarian

**Affiliations:** 1Institute of Systems Biology (INBIOSIS), Universiti Kebangsaan Malaysia, Bangi 43600, Selangor, Malaysia; 2Faculty of Bioeconomic & Health Sciences, Universiti Geomatika Malaysia (UGM), Kuala Lumpur 54200, Selangor, Malaysia; 3UKM Molecular Biology Institute (UMBI), UKM Medical Center, Kuala Lumpur 56000, Selangor, Malaysia; 4Laboratory of Cancer Research UPM-MAKNA (CANRES), Institute of Bioscience, Universiti Putra Malaysia, Serdang 43400, Selangor, Malaysia; 5Drug Discovery and Synthetic Chemistry Research Group, Department of Pharmaceutical Chemistry, Kulliyyah of Pharmacy, International Islamic University Malaysia, Kuantan 25200, Pahang, Malaysia; 6Kulliyyah of Allied Health Sciences, International Islamic University Malaysia, Kuantan 25200, Pahang, Malaysia

**Keywords:** wound healing, flavonoids, pathways, signaling, scar, natural products

## Abstract

Wounds are considered to be a serious problem that affects the healthcare sector in many countries, primarily due to diabetes and obesity. Wounds become worse because of unhealthy lifestyles and habits. Wound healing is a complicated physiological process that is essential for restoring the epithelial barrier after an injury. Numerous studies have reported that flavonoids possess wound-healing properties due to their well-acclaimed anti-inflammatory, angiogenesis, re-epithelialization, and antioxidant effects. They have been shown to be able to act on the wound-healing process via expression of biomarkers respective to the pathways that mainly include Wnt/β-catenin, Hippo, Transforming Growth Factor-beta (TGF-β), Hedgehog, c-Jun N-Terminal Kinase (JNK), NF-E2-related factor 2/antioxidant responsive element (Nrf2/ARE), Nuclear Factor Kappa B (NF-κB), MAPK/ERK, Ras/Raf/MEK/ERK, phosphatidylinositol 3-kinase (PI3K)/Akt, Nitric oxide (NO) pathways, etc. Hence, we have compiled existing evidence on the manipulation of flavonoids towards achieving skin wound healing, together with current limitations and future perspectives in support of these polyphenolic compounds as safe wound-healing agents, in this review.

## 1. Introduction

The skin is the largest organ in the human body in terms of surface area. It protects internal tissues from mechanical damage, microbial infection, UV light, and extreme temperatures [1]. A wound is an injury to the integument or underlying structures; it is the visible outcome of individual cell death or damage, which may or may not result in a loss of skin integrity, impairing the tissue’s physiological function. When the natural structure and function of the skin are damaged, injuries occur [2]. When an injury occurs on the skin, there are complex interactions involving different cell types, cytokines, mediators, and the vascular system as parts of a natural physiological reaction that occur in wound healing.

Wounds are classified as either acute or chronic based on the root causes and effects of each. An acute wound is on in which there is a quick loss of skin integrity, which is frequently due to trauma or surgery. Any surgical wound that heals by primary intention is considered an acute wound, as is any traumatic or surgical wound that heals by secondary intention. An acute wound is anticipated to proceed through the stages of typical healing, leading to the wound’s closure. Although a chronic wound is a wound that does not heal in a timely and orderly manner, resulting in anatomic and functional integrity loss. A chronic wound is “frozen” in the inflammatory phase for longer than the typical estimated healing time (4 weeks) and does not heal or react to treatment. This pathologic inflammation results from a delayed, ineffective, or disorganized healing process. Both intrinsic and extrinsic variables, such as drugs, poor diet, comorbidities, or incorrect dressing options, can cause a wound to take longer to recover [3].

To further comprehend chronic wounds, the three types of these injuries include pressure ulcers, diabetic ulcers, and vascular ulcers (such as venous and arterial ulcers). Each of these wounds has some commonalities, such as persistent infections, prolonged or severe inflammation, the development of drug-resistant microbial biofilms, and the incapacity of dermal and/or epidermal cells to react to reparative stimuli [4]. It is understood that pressure ulcers can develop over pressure points, such as on the heel of a bedridden patient or on the side of the foot from wearing tight shoes. Conventionally, any ulcer in a diabetic patient is interpreted as a diabetic ulcer. However, vascular reflux is most likely the cause of ulcers that have developed at the ankle, calf, or pretibial sites [5]. As for an acute wound, it is a skin injury that occurs suddenly rather than gradually. It heals at the normal wound-healing rate, which is predictable and expected. Acute wounds can occur anywhere on the body and range in severity from minor scratches to deep wounds that damage blood vessels, nerves, and muscles [6].

According to various research reports and data published by Mission Regional Medical Center in 2020, it was estimated that around 6.7 million people in the world were suffering from chronic wounds. The global chronic wound care market is projected to grow from USD 12.36 billion in 2022 to USD 19.52 billion by 2029, at a CAGR of 6.7% in the forecast period [7]. On a slightly different note, treatment for wounds that cause scarring is as crucial as chronic wounds. This is due to the immense numbers of patients, which is estimated to be around 100 million patients per year in the developed world, that were affected by scars from 55 million elective surgeries and 25 million trauma-related surgeries [8].

The hunt for the best wound-healing treatment has long been a goal of research on wound-healing therapy including surgical, non-surgical, and drug treatments. With an emphasis on wound-healing drugs, commonly used drugs are non-steroidal anti-inflammatory drugs (NSAIDs) such as ibuprofen, warfarin, colchicine, aspirin, (antiplatelets), prednisolone (corticosteroids), heparin (anticoagulants), nicotine, cocaine, and adrenaline (vasoconstrictors). However, there are some drawbacks from these drugs; for example, vasoconstrictors cause tissue hypoxia by adversely affecting the microcirculation, leading to impaired wound healing. Vasoconstrictors such as nicotine, cocaine, adrenaline (epinephrine), and ergotamine should be avoided in patients with acute, surgical, or chronic wounds [9]. In the early 19th century, nicotine and cocaine were both used medicinally, but cocaine was outlawed in the US in 1914 due to its hallucinogenic properties. Additionally, nicotine has been highlighted as a potential contributor to pharmaceutical overreliance [10,11]. In 1808, ergotamine was widely used to precipitate childbirth and to control post-partum hemorrhage due to its remarkable uterotonic and vasoconstrictor effects [12]. The European Medicines Agency’s Committee for Medicinal Products for Human Use (CHMP) has suggested prohibiting the use of medications that include ergot derivatives. Because the hazards outweigh the benefits in these indications, these medications should no longer be used to treat a variety of illnesses involving blood circulation issues.

Therefore, natural-product-based treatments have been naturally explored for their ability to serve the optimum effects of wound healing to patients accordingly. Flavonoids, which are prominently known for their wound-healing properties, have recently been reported to be implemented in numerous formulations, topical ointments, and dressings for wound healing [13,14]. To date, flavonoids for wound healing have been thoroughly discussed and reported through various pathways such as Wnt/β-catenin, Hippo, TGF-β, Hedgehog, JNK, (Nrf2/ARE, NF-κB, MAPK/ERK, Ras/Raf/MEK/ERK, PI3K/Akt, NO, etc. [15]. Thus, this review sets out to establish knowledge and thoroughly discuss the flavonoids and their underlying mechanisms in wound-healing treatments.

## 2. Wound Healing

Wound healing is a multi-phased process that occurs when the normal anatomical structure and function of skin tissues are disrupted. Inflammation, granulation, wound shrinking, collagen creation, epithelial closure, and scar formation are all part of the process. The smoothness of these phases promotes wound healing and restores the skin’s previously compromised anatomical condition and function [16]. Wound healing is a four-phase process that involves a complicated series of reactions and interactions between cells and mediators; these phases are hemostasis, inflammation (0–3 days), proliferation (3–24 days), and maturity (24 to 365 days). Figure 1 depicts the physiological wound-healing phases. These phases and their physiological functions must occur in the correct order, at a specified moment, and at an optimal intensity for a specific length. In the first phase (bleeding and hemostasis), hemostasis occurs immediately after bleeding and reduces blood flow by constricting blood vessels to stop bleeding following vascular injury. The phenomenon that subsequently occurs is the aggregation of platelets, degranulation, and the production of fibrin, which is also known as thrombus.

However, in the second phase (inflammation), the events are the release of cytokines, growth factors, infiltration of lymphocytes, neutrophils, monocytes, and their differentiation into macrophages in the area, causing hemostasis and acute inflammation in the area of a wound. The third phase is proliferation; the occurrences are increases in the keratinocyte, fibroblast, and endothelial and extracellular matrix (ECM) production; re-epithelialization; angiogenesis; collagen synthesis; and leukocyte migration and proliferation in the wound area. The last phase serves for remodeling and maturity in that the stretching and pulling forces interact with the scar tissue. Meanwhile, vascular maturation, regression, and collagen remodeling occur, thus helping collagen regain the normal alignment needed [17].

In more detail, the initial phase of hemostasis begins with vascular constriction and fibrin clot formation shortly after injury. Pro-inflammatory cytokines and growth factors such as transforming growth factor (TGF), platelet-derived growth factor (PDGF), fibroblast growth factor (FGF), and epidermal growth factor are released by the clot and surrounding wound tissue (EGF). After the bleeding has been stopped, inflammatory cells migrate into the wound (chemotaxis) and initiate the inflammatory phase, which is marked by the successive infiltration of neutrophils, macrophages, and lymphocytes [18]. Neutrophils are important for clearing invading microorganisms and cellular debris from wounds, but they also produce chemicals such as proteases and reactive oxygen species (ROS), which can cause extra bystander damage.

Meanwhile, macrophages are involved in wound healing in a variety of ways. Macrophages release cytokines in the early stages of a wound, which increase the inflammatory response by attracting and activating extra leukocytes. Macrophages are also responsible for generating and removing apoptotic cells (such as neutrophils), allowing inflammation to be resolved. As macrophages remove apoptotic cells, they undergo a phenotypic shift to a reparative state, which encourages keratinocytes, fibroblasts, and angiogenesis to aid tissue regeneration. In this way, macrophages facilitate the transition to the proliferative phase [19]. The proliferative phase is characterized by epithelial proliferation and migration across the provisional matrix within the wound, and it usually follows and overlaps with the inflammatory phase (re-epithelialization). The most significant cell types found in the reparative dermis are fibroblasts and endothelial cells, which support capillary expansion, collagen synthesis, and the formation of granulation tissue at the site of damage. Fibroblasts in the wound bed create collagen, glycosaminoglycans, and proteoglycans, all of which are important components of the extracellular matrix (ECM). Wound healing reaches the ultimate remodeling phase after vigorous proliferation and ECM synthesis, which can continue for years. During this phase, many of the newly formed capillaries regress, and the wound’s vascular density returns to normal. One of the most important aspects of the remodeling phase is ECM remodeling to resemble normal tissue architecture. The wound physically shrinks during the healing process, which is thought to be mediated by contractile fibroblasts (myofibroblasts) that develop in the injury [18,20].

Wound healing is a burden and a cost to a country’s healthcare system, potentially arising because of diabetic ulcers or accidents, as opposed to cuts that occur on a routine basis. According to reports, more than 13 million people worldwide experience chronic wounds each year, and the number of patients affected is steadily growing as the world’s population ages [21]. Chronic wounds of the lower leg are notoriously difficult to heal, resulting in decreased patient satisfaction, increased morbidity and mortality, and a significant rise in healthcare costs [22]. The situation worsened during the coronavirus (COVID-19) pandemic, which wreaked havoc on worldwide healthcare, particularly wound care [23]. Furthermore, the wound-healing process might also slow down due to systemic factors such as poor sleep habits, poor nutrition, less exercise, and a high predilection for the use of alcohol, cigarettes, and other drugs. Aside from systemic factors, local factors, such as oxygenation, infection, foreign bodies, and venous sufficiency, might also cause wound-healing problems. All of these variables may contribute to the slowing of the healing process [17,24]. On a positive note, extensive studies have been conducted on exploring the potential of natural compounds with anti-inflammatory, antioxidant, anti-bacterial, and pro-collagen production capabilities as wound-healing agents. The presence of bioactive phytochemicals such as alkaloids, essential oils, flavonoids, tannins, saponins, and phenolic compounds in natural resources makes it possible to exploit all of their bioactive qualities [16]. Several studies employing medicinal plants have recently discussed the creation of new resources and technologies with the potential to heal a variety of acute and chronic wounds with minimum side effects, ease of administration, increased efficacy, and cheaper treatment costs for patients [25,26,27] Flavonoids are one of the most important and promising families of natural compounds for treating skin lesions [28,29].

Regrettably, the management of both acute and chronic wounds continues to be one of the most challenging healthcare issues in the world. Wounds of all forms, including burns, continue to be a major public health concern in both developing and industrialized countries [30]. Additionally, poor lifestyle practices and unhealthy choice of nutrition falters wound management. From a more positive perspective, nano dressing, negative pressure, medicinal plants, synthetic polymers, gene therapy, stem cells, growth factors, and functionalized silk biomaterials are some of the various wound-healing techniques that can help reduce the cost of wound care. Flavonoids, which are secondary metabolites present in several types of vegetables and fruits, are frequently reported in the literature as being responsible for numerous in vitro and in vivo biological activities, including the treatment of wounds [14]. Hence, this review aims to understand the effect of flavonoids on wound healing through previously reported pathways.

## 3. Current Treatment for Treating Wound Healing

The art of wound care has been practiced traditionally since time immemorial; however, given the current advances in recent knowledge, few drugs have improved wound healing. The search for an effective route of treatment with minimal side effects remains elusive due to the complexity of tissue restoration. Chronic and acute wounds are the most prevalent of all injuries, affecting people worldwide with no effective treatments. Despite substantial progress in wound care over the last few decades, vigorous breakthrough studies continue to search for innovative therapeutic techniques that target acute and chronic wound management [31].

Chronic wounds are commonly known as difficult-to-heal injuries with a physiological impairment of tissue restoration and typically do not improve according to the timely sequence of tissue repair [32]. All injuries are susceptible to becoming chronic wounds and are etiologically divided into four groups, i.e., pressure, diabetic, venous, and arterial ulcers. These wounds typically exude pus and odors, and applied dressings are always prominent. Microbial colonies such as *Staphylococcus aureus* and *Pseudomonas aeruginosa* often invade chronic wounds, forming biofilms and leading to critical infections as well as impacting the overall process of wound repair [33]. They secrete endo- and exotoxins while modulating several immune responses at the wound site, producing excessive proteases that could degrade the extracellular matrix and impede the overall process of wound healing [34,35].

On the other hand, acute wounds are associated with the external destruction of intact skin and are composed mainly of burns, cuts, and bites, as well as traumatic lacerations [36]. The treatment regime depends on the type, severity, and size of the wounds. Similar to chronic wounds, acute wounds are also susceptible to microbial infections, with an estimated microbial colonization of 5 to 26% [37]. Burn wound infections encompass more than half of mortalities per year, and a number of pathogenic microbes such as *S. aureus*, *P. aeruginosa*, and *Escherichia coli*, as well as coagulase-negative *Staphylococci*, have been identified in post-burn injuries [38]. Of all, there are no current effective treatments for both wounds in terms of side effects and cost effectiveness. Figure 2 illustrates the current treatment utilized for treating acute and chronic wounds. Thus, in this section, current treatments available in wound healing are thoroughly discussed.

### 3.1. Wound Dressings

Wound dressing is typically used to cover a specific site of injury from microorganisms. It acts as an insulator by moisturizing the wound site and promoting tissue granulation and epithelialization [39]. The chief goal of wound dressing is to minimize the risk of pathogenic infections and to reduce scarring. Myriads of wound dressings, such as medicated, traditional, and modern dressings, have been developed, each promoting flexibility and oxygen exchange at the wound site. Table 1 highlights the distinct types of wound dressings, their functions, and their limitations.

### 3.2. Antibiotics

Antibiotics were first discovered in the early 19th century by Alexander Fleming and have been broadly utilized ever since to combat infections. A plethora of antibiotics, namely penicillin, streptomycin, tetracycline, and vancomycin, have been developed to treat specific pathogenic infections since their discovery. In wound healing, topical applications such as creams, pastes, gels, and lotions are desirable as treatment because they are more localized and provide better drug bioavailability. Topical antibiotics such as neomycin, povidone-iodine, and silver sulfur diazine are extensively prescribed for their effectiveness to prevent infections [49]. However, the prolonged use of topical and systemic antibiotics on recovered wounds has resulted in major antibiotic resistance worldwide [50]. Recent data reported that bacterial infections related to wound healing have been reaching rates of antibiotic resistance of almost 70%, which could cause the spreading of resistant strains [35]. In view of this, more than half a million deaths have been recorded annually, and deaths are forecasted to increase by 10 million per year by 2050 [51]. In addition, the use of topical antibiotics may also cause hypersensitivity and allergic reactions [52]. Alternatively, antiseptics such as octenidine hydrochloride, polyhexamethylene biguanide, and sodium hypochlorite are preferred by clinicians, as these medications have potent anti-microbial properties as well as anti-biofilm-forming activity [53].

### 3.3. Surgical Methods

Debridement is one of the common techniques used to treat acute wounds or injuries such as burns, as it allows wounds to advance from the inflammatory to the healing phase. It facilitates wound epithelialization and minimizes the risk of bacterial infections by removing necrotic tissues, senescent cells, and biofilms from the wound site. However, biofilm removal is a laborious process due to its adhesive properties at the wound site, and the bioavailability of specific antibiotics are usually very low [54]. In the case of deep tissue infection, i.e., that involving muscle or adipose layer, amputation is the best approach for sepsis prevention [55]. Nonetheless, the amputation approach does not resolve the risk of reinfection on patients who have diabetic foot ulcers, as they are still susceptible to a 40% mortality rate after amputation [56].

To date, no known approach has been identified for its successful wound-healing activity and its ability to initiate distinct molecular and cellular mechanisms in various types of wounds. Previously mentioned treatments not only have side effects but are also expensive. Hence, an alternative treatment with minimal side effects is warranted. In fact, recent studies have suggested that natural products could be used to treat wounds. A mixture of black seed oil and honey has been reported to promote wound healing without causing cellular toxicity [57]. Honey in gellan gum (GG) hydrogel containing virgin coconut oil (VCO) was shown to hasten the healing process of an injury [58]. Additionally, there is preliminary evidence that granulated sugar is an effective wound cleanser and is safe to use in patients with insulin-dependent diabetes [59].

Current trends have been focusing on herbal medicines for their low side effects as well as drug resistance in treating wound infections [55]. Herbal medicines and herbally derived substances have been used in folklore medicine since time immemorial as they exert multiple therapeutic benefits, for instance, anti-inflammatory and antioxidant activities and modulation of cell proliferation [60]. The therapeutic properties of herbal medicine are unique to the mixture of bioactive components. Among them, flavonoids are one of the promising key bioactive ingredients for wound-healing properties [15]. Their structure–activity relationship (SAR), i.e., the presence of a hydroxyl group is essential for various biological activities, for example, anti-bacterial, antioxidant, and anti-inflammatory activities [61,62].

## 4. Socioeconomic Effects of Morbidity and Mortality from Chronic Wounds

The number of people with chronic wounds has been rising like a “silent epidemic” [63]. Chronic wounds account for 2–4% of the health expenditure in Western nations [64]. This has frequently led to inadequate planning and implementation of preventative, treatment, and management methods. Therefore, wound research is an important yet undervalued topic of study [22]. The ideal plan for wound management includes exploring ways to minimize economic strain while simultaneously reducing morbidity and mortality. Some chronic wounds take years or even decades to heal. Patients may experience extreme pain, major mental and physical discomfort, decreased mobility, and social isolation during this period [65]. Reduced mobility can impair a person’s ability to work, complete household chores, and care for one’s personal hygiene. Mobility limitations are one of the worst aspects of having a wound, according to patients, as these limitations can impede independence and quality of life. The loss of independence caused by functional deterioration and social isolation has a negative impact on overall health and wellbeing [66].

Chronic wounds place a significant financial strain on healthcare systems, in terms of both direct and indirect expenditures. The frequency of dressing changes, the length of hospital stays for treatment, the frequency of complications, and, in particular, the time spent on healthcare all contribute significantly to the expense of wound care [67]. It will be very helpful to create new, low-cost therapeutic and preventative technologies to meet the needs of a specific healthcare environment, especially in low- and middle-income nations in which a lack of access to affordable, high-quality healthcare is a grave problem. In addition, hospitals also need to have wound registries because they will serve as a trustworthy data collection tool. Wound registries can be utilized as continual quality-improvement tools or to standardize wound observation. These registries will aid in the development of wound care techniques required in healthcare facilities including hospitals and nursing homes [66].

## 5. The Effect of Flavonoids on Wound Healing

Flavonoids are generally recognized as the most common bioactive compounds on the planet. Flavonoids are abundant in fruits and vegetables. Flavonoids are phenolic compounds that can be found in fruits, vegetables, herbs, cocoa, chocolate, tea, soy, red wine, and other plant food and beverage products. Flavonoids are composed of two aromatic rings (A and B rings) connected by a three-carbon chain to produce an oxygenated heterocyclic ring (C ring). Based on changes in the general structure of the C ring, functional groups on the rings, and the position at which the B ring is linked to the C ring, flavonoids can be classified into various classes such as flavones, flavonols, flavanols, flavanones, isoflavones, anthocyanins, and chalcones. Figure 3 represents the basic skeletal structure of flavonoids and their classes as well as the examples. Individual compounds within each category are distinguished by distinct hydroxylation and conjugation patterns [68,69,70]. Flavonoids are one of the most attractive and promising families of natural products for treating skin problems. The structure–activity connection (SAR) of flavonoids is one of the key elements that lead to this feature. The presence of hydroxyl groups in their chemical structure, particularly at positions 5, 7, 3, and 4, is critical for their antibacterial, antifibrotic, antioxidant, and anti-inflammatory effects due to their high hydroxylation levels [61].

Catechins (flavan-3-ol), which modulate wound healing, are one of the most widely studied flavonoids [71]. Some researchers hypothesized that flavonoids such as apigenin could help treat skin injuries by inhibiting fibroblast development, because wound healing was delayed due to insufficient or excessive fibroblast operation. Lutein is a prominent dietary flavonoid that can be found in a variety of medicinal plants, as well as ordinary vegetables and fruits. It has also been used as a wound healer in a variety of wound models [72]. Rutin (quercetin-3-O-rutinoside), which is found in many medicinal plants, has wound-healing properties [73].

Flavonoids protect body cells from oxidative damage, which can cause disease, and they have advantageous defensive actions. Flavonoids such as quercetin form *o*-quinones, which, upon treatment with a solvent containing water, restored the potent antioxidant activity of the quinone. For example, carnosol quinone is the antioxidation product of carnosol, which possesses a very weak antioxidant activity. Quinones are a class of toxicological intermediates that can cause a number of harmful consequences in vivo to cells, including acute immunotoxicity, cytotoxicity, and carcinogenesis. In contrast, quinones can provide cytoprotection by inducing detoxifying enzymes, anti-inflammatory actions, and altering the redox state. The methods by which quinones exert these actions can be rather complicated [74].

Many flavonoids have been characterized as potent reactive oxygen species (ROS) inhibitors, making them vital antioxidant food components. The influence of ROS on the oxidation of quercetin, kaempferol, morin, catechin, and naringenin was investigated. The reaction rates determined by spectrophotometry and oxygen consumption were drastically different. Quercetin possesses powerful antioxidant and anti-inflammatory effects, which support its potential use in wound healing. Additionally, quercetin can reduce both acute and chronic inflammatory stages. Reactive oxygen species and oxidative stress have only a minor role in the normal physiology of wound healing, but an excess of either can hinder healing. The use of antioxidants as most flavonoids is thought to speed up wound healing by reducing oxidative stress in the wound [74]. The list of potential flavonoids requires more research to promote more discovery.

### 5.1. Efficacy Evidence of Flavonoid as a Wound Healing Agent

Hemostasis, inflammation, proliferation, and remodeling are the sequential steps of wound healing. Each of these phases is governed by distinct growth factors, mediators, immune cells, and metabolites within signaling pathways that promote wound-healing activity. Numerous in vitro and in vivo investigations have demonstrated that flavonoids possess wound-healing effects. To determine the efficacy of flavonoids in wound healing, it is customary to detect cell migration in a scratch assay using a cell line study in accordance with the proper dosage of the tested flavonoids. Typically, the reliability of the test is determined by comparing treated and untreated cells. The inability of cell line research to simulate the actual process of wound healing, which involves inflammation and the immune system, necessitates subsequent in vivo testing on an organism [75]. Table 2 summarizes the effect of flavonoid treatment in in vivo and in vitro testing.

Previous research has demonstrated the efficacy of flavonoids in shortening the period of wound healing by influencing collagen breakdown and MMP-2 activity following 24-h therapy [76]. The wound-healing rate was increased by 51% after being treated with quercetin-3-oleate at its highest concentration of 1 μM, with slight TGF-β production and MMP-9 release [77]. Although the upregulation of TGF-β was significantly induced, the absence of MMP-9 in HaCaT cell cultures suggests that the wound-healing capacity of these cells may be governed by distinct signaling pathways. Several studies with various in vivo testing sources and methodologies have indicated delayed wound healing. This issue is currently a major concern for healthcare professionals and individuals globally [78]. Hesperidin, one of the flavone glycosides, was, therefore, tested for its effects on diabetic foot ulcers. Compared to the control mice, the rate of wound closure for a chronic diabetic foot ulcer was less than 21 days. In addition to accelerating angiogenesis by elevating the expression of VEGF-c, Ang-1/Tie-2, TGF-, and Smad-2/3 mRNA, a considerable increase in wound closure has been observed [79]. Angiogenesis is a crucial process that promotes the regeneration of new tissue and organ development while simultaneously supplying wounds with nutrients and new cells [80,81]. Additionally, the rate of epithelialization promoted by flavonoids is assisted by their other antimicrobial properties [82].

In addition, it is essential to evaluate wound closure as a role in wound-healing activity by examining the wound margins until they heal. Consequently, in another trial involving a hydrogel containing flavonoid glycoside, a reduction in wound area over a period of 4 to 16 days was observed [83]. This discovery was made in several mouse models using different medication formulations denoted as H1 (0.0020% *w*/*w*), H2 (0.0025% *w*/*w*), and H3 (0.0030% *w*/*w*), with H2 proving to be the most effective formulation for encouraging wound closure beginning on day 4. In accordance, the flavonoid has played a significant part in the restoration of bone abnormalities, which has become one of the obstacles of clinical therapy [84]. The investigation was evaluated on 44-week-old mice that were placed into groups of eight and administered varying amounts of flavonoids. The therapeutic activity was discovered to be directly related to the higher dosage, i.e., 100 mg/kg per day was found to be the optimal dosage which can provide successful treatment for bone defects. Flavonoid is capable of initiating osteoblast differentiation, hence promoting angiogenesis and indirectly reflecting the activity of proliferation. In features of the veterinary application, an extract with a high flavonoid content from *Libidibia ferrea* has been suggested to be commercialized because histologically, it displays substantial numbers of fibroblasts, newly created capillaries, and collagen fibers from a concentration of only 5% [85].

In a separate study, quercetin-loaded liposomal hydrogel was evaluated for its ability to stimulate cell proliferation both in vitro and in vivo [72]. The multiphase system was successfully formulated, and its in vivo test yielded considerable results. As compared to the control group, wound excision in rats treated with quercetin-loaded liposomes resulted in 52.26% more wound contractions within four days of treatment. The authors highlighted that the produced hydrogel’s combination of excellent hemocompatibility, intact mechanical strength, and low swelling ratio led to quick wound closure. Quercetin has also been reported to demonstrate the ability to reduce fibrosis and scar formation during wound healing, promote fibroblast cell proliferation, reduce immune cell infiltration, and trigger alterations in fibrosis-related signaling pathways [86].

Reportedly, diabetic people typically have healing difficulties. Researchers have published several studies evaluating a suitable source for accelerating wound healing in diabetics. In a previous study [87], the rate of wound healing in incisional and excisional wounds of diabetic rat models was investigated. The wound was treated for fourteen days with remarkable results. In diabetic rats with excisional wounds, the rate of healing was determined to be 92.12%. Reepithelization scores were measured via histological examination on the 7th and 14th days, and they were found to be greater than the control alongside other wound-healing characteristics including granulation, angiogenesis, and inflammation. The need for these characteristics to produce a meaningful influence on the wound defines the efficacy of a particular studied drug, which in this case appears to be kaempferol, a potent wound-healing agent.

### 5.2. Antibacterial Properties of Flavonoids in Wound Healing

The skin microbiota consists of microorganisms that play a pivotal role against pathogens. As injury occurs, microorganisms such as commensal bacteria will colonize the wound and become pathogenic as they produce biofilms (Figure 4) [88]. *Staphylococci* families are among the first colonizers, as they are major inhabitants of our skin and mucous membranes [89]. Similar to any other injury, acute and chronic wounds are at risk of bacterial infections that could be life threatening and could cause multiple inflammations in our body systems [90]. Bacterial infections are one of the major contributing factors for delayed wound healing and are susceptible to new infections and treatment failures, as they accumulate excessive ROS. At a low level, ROS promotes angiogenesis, promoting blood perfusion at the affected area [90]. Nonetheless, excessive ROS is the causative agent that contributes to biofilm production which could prevent the transitional stage of the inflammatory process towards the proliferative phase [91]. Thus, the search for a safe anti-bacterial agent is warranted. Among all of the safe anti-bacterial agents, flavonoids have been evaluated for their anti-bacterial properties owing to their accessibility and non-toxic therapeutic use. Moreover, a growing interest in flavonoids and terpenes for their anti-bacterial properties has been reported [92]. Therefore, the therapeutic use of flavonoids as anti-bacterial agents on wound healing has been highlighted with examples (Table 3) in this section.

## 6. Pathways Involved in Wound Healing

Numerous studies have suggested that flavonoids have wound-healing abilities; hence, this section will address the various widely accepted, reported mechanisms in particular (Table 4).

### 6.1. Wnt/β-Catenin Pathway

The Wnt/β-catenin signaling pathway is a pathway that is essential for embryological growth and renewal of mature tissue homeostasis. The Wnt/β-catenin pathway plays crucial roles in numerous wound-healing processes involving tissue remodeling and cell growth. Additionally, it plays a role in the expression of growth factors and stem cell activation and improves wound angiogenesis. In fact, there are numerous reports that the Wnt family is involved in biological processes such as cell proliferation, apoptosis, differentiation, and the maintenance of pluripotency in stem cells [111,112]. According to reports, the flavonoids that contributed towards wound healing were quercetin and berberin [98,113].

### 6.2. Hippo Pathway

The Hippo pathway is an evolutionarily conserved signaling mechanism that holds important functions in tissue regeneration, immunological regulation, epithelial homeostasis, organ development, and wound healing. Many of these functions are carried out by the transcriptional effectors YAP and TAZ, which regulate the TEAD family of transcription factors to control gene expression [114]. According to previously published research, it has been found that there is an interaction between the YAP/TAZ (Hippo pathway) and TGF-1/Smad signaling pathways during the healing process, which suggests that the findings might have pleiotropic effects that affect collagen production, cell growth, and wound healing [115]. Despite that, there are hardly any reports on flavonoids’ beneficial effects on wound healing.

### 6.3. Transforming Growth Factor β (TGF-β) Pathway

The TGF-β signaling pathway controls a number of biological functions, such as cell division, proliferation, death, plasticity, and migration [116]. This pathway is associated with a variety of wound-healing processes, including inflammation, promotion of angiogenesis, boosting of fibroblast growth, collagen synthesis and deposition, and remodeling of the new extracellular matrix [117]. Hesperidin, quercetin, glycitin, naringin, and genistein are flavonoids that have been linked to the healing of wounds via this pathway [15].

### 6.4. Hedgehog Pathway

The Hedgehog signaling route is a signaling pathway that provides embryonic cells with the information they need to differentiate properly. This pathway is vital for healthy embryonic development and is crucial for the maintenance, renewal, and regeneration of adult tissue [118]. The results from an investigation conducted by Le et al. (2008) [107] strongly suggested that Shh signaling is involved in the natural healing of regular, definite, full-thickness wounds [119], thus implying that this pathway has potential for wound healing. However, there are no flavonoids reported so far for wound healing through this pathway.

### 6.5. Jun N-Terminal Kinase (JNK) Pathway

Jun N-terminal kinase (JNK), also called stress-activated protein kinase (SAPK), is one of the three main members of the mitogen-activated protein kinase (MAPK) superfamily. The other two are extracellular signal-regulated kinase (ERK) and p38 MAP kinase [120]. A few studies have implicated the JNK pathway in the regulation of cell migration. Cell migration is required for wound healing. Cell migration can be separated into multi-step cyclic processes. The fundamental migratory cycle consists of the elongation of a protrusion, the creation of stable attachments along the protrusion’s leading edge, the translocation of the cell body forward, the release of adhesions, and the contraction of the cell’s rear. In general, the JNK pathway is a “death” signaling mechanism. It regulates the cell’s response to damaging extracellular stimuli such as inflammatory cytokines, UV-irradiation, and gamma-irradiation, among others. Under the influence of these noxious stimuli, DNA may undergo mutation or damage. In the situation in which DNA damage cannot be repaired promptly, the cell must be programmed to die (also known as apoptosis) to prevent further mutation or harm. In a wound-healing assay, JNK-inhibited cells had a lower migration rate than control cells after 12 h in the presence of basic fibroblast growth factor (bFGF) [121].

The JNK pathway is important for the healing of imaginal disc wounds, as it has been shown to be in other types of wounds in Drosophila, including embryonic dorsal closure, thoracic closure, and adult epithelial wounds [122]. The antioxidant and signaling characteristics of flavonoids have been linked to their physiological effects [123]. The green tea polyphenol (−)-epicatechin-3-gallate (ECG) effectively protects HaCaT keratinocytes from ultraviolet B (UVB)-induced damage. The keratinocytes are protected by ECG from oxidative stress caused by H_2_O_2_ and photodamage caused by UVB, likely through blocking the activation of the JNK signaling pathway [124]. Quercetin prevents apoptosis and increases cell viability by inhibiting oxidant-induced signaling in the JNK and p38 MAPK (mitogen-activated protein kinase) pathways [125]. Citrus flavonoids have been shown to increase pro-survival signaling molecules, such as Akt/protein kinase B and p38 mitogen-activated protein kinase, and prevent the expression of JNK, which induces apoptosis [123]. Enhancing cell survival requires the activation of ERK, Akt/PKB, PI3K, and PKC, whereas avoiding apoptosis requires the downregulation of P38 and JNK.

### 6.6. Nuclear Factor Erythroid 2-Related Factor 2/Antioxidant Response Element (Nrf2/ARE) and Nuclear Factor-κB (NF-κB) Pathways

Wound healing is promoted by nuclear factor erythroid 2-related factor 2 (Nrf2) and nuclear factor kappa B (NF-κB) transcription factors via their anti-inflammatory and antioxidant effects or via the immune system [126]. The primary regulator of intracellular redox homeostasis is (Nrf2), a redox sensitive transcription factor. It drives the expression of cytoprotective genes and boosts the generation of antioxidants that scavenge free radicals. In numerous pathophysiological diseases—including diabetes and its resulting conditions such as diabetic foot ulcers, chronic kidney disease, and diabetic nephropathy—activators of Nrf2 have been observed to reduce oxidative stress and improve the process of wound healing. Chronic wounds take longer to heal due to a number of different factors, including reactive oxygen species (ROS), systemic illness, trauma, immune suppression, ischemia, an imbalance of pro/anti-inflammatory cytokines, interleukins, leukotrienes, and complement factors at the wound site, and a lack of extracellular matrix proteins (ECM). Wound healing is promoted by Nrf2 and NF-κB transcription factors via their anti-inflammatory and antioxidant effects or via the immune system. In wound healing, Nfr2 and Nf-κB perform a key and reciprocal role. Nrf2 regulates repair-related inflammation and protects against excessive accumulation of ROS, whereas Nf-κB activates the innate immune response, cell proliferation, and cell migration and modulates the expression of matrix metalloproteinases, secretion, and stability of cytokines and growth factors for wound healing [126].

Through activation of Nrf2, bioactive compounds, both those that come from nature and those that are made in a lab, can help diabetic wounds heal in important ways [127]. The role of Nrf2 in the wound-healing process has been a focus of interest for therapeutic research. In the presence of severe tissue injury and ROS generation, Nrf2 prevents the activation of cytoprotective genes that leads to apoptosis of keratinocytes [128]. It is widely acknowledged that Nrf2 functions as a defense signal under oxidative stress and protects cells by decreasing ROS (Figure 5).

It has been shown that some bioactive compounds reduce cellular stress and consequently accelerate cell proliferation, neovascularization, and healing of injured tissues by activating Nrf2 expression. During wound healing, the Nrf2 pathway protects against oxidative stress via the production of antioxidative enzymes [128]. Injuries to the nasal mucosa that persist after nasal trauma, septoplasty, turbinate treatment, functional endoscopic nasal surgery, and tumor removal can be treated with curcumin because curcumin significantly improves wound healing. Curcumin dramatically speeds up the healing process by reducing inflammation. In the early stages of wound healing, curcumin has an apoptotic effect [129,130]. Some of the flavonoids discussed in this review were capable of increasing levels of SOD, CAT, GSH, GST, and GPX, all of which point to an antioxidant effect, and they also helped the body’s natural healing process.

NF-κB is a transcription factor that regulates protein kinases, regulates cell activation and proliferation directly, and enhances the expression of several pro-inflammatory genes in cells [131]. Flavonoids inhibited the expression of nuclear factor kappa B (NF-kB) and reduced the levels of inflammatory mediators such as prostaglandin E2 (PGE2), leukotriene B4 (LTB-4), interleukin 1 (IL-1), tumor necrosis factor (TNF-), interleukin 6 (IL-6), and interferon (IFN-) [88]. Supplementation with luteolin significantly reduced protein expressions of inflammatory factors such as matrix metalloproteinase (MMP)-9, tumor necrosis factor (TNF)-, interleukin (IL)-6, and IL1-. Additionally, luteolin downregulated nuclear factor (NF)-B while also inducing increases in anti-oxidative enzymes such as superoxide dismutase 1 (SOD1) and glutathione peroxidase (GSH-Px) [132].

### 6.7. Mitogen-Activated Protein Kinase/Extracellular Signal-Regulated Kinase (MAPK/ERK) and Phosphatidylinositol 3-Kinase (PI3K)/Protein Kinase B (P13/AKT) Pathway

Mitogen-activated protein kinase/extracellular signal-regulated kinase (MAPK/ERK) signaling is heavily involved in the regulation of cell migration and proliferation. Activation of the MAPK/ERK signaling pathway is a significant regulator of the migration of different cell types [133]. The MAPK pathways have long been acknowledged in the scientific literature for their roles in angiogenesis during wound healing [134]. Quercetin may be a bioactive substance that can help with the symptoms of atopic dermatitis. It has anti-inflammatory and antioxidant properties and speeds up wound healing through the ERK1/2 MAPK [135]. Vaccarin is a flavonoid glycoside with several biological roles. Vaccarin promotes wound healing and the proliferation of endothelial cells and fibroblasts at the wound site. A previous study showed that vaccarin can increase the expressions of p-Akt, p-Erk, and p-bFGFR to cause angiogenesis and speed up wound healing in vivo. The MAPK and PI3K/AKT signaling pathways are known to play a role in angiogenesis throughout wound healing [134]. The MAPK/ERK and PI3K/AKT signaling pathways control this process [136]. The PI3K/Akt signaling system controls cell proliferation, differentiation, and migration in addition to controlling angiogenesis and metabolism. Additionally, it may facilitate skin growth and homeostasis. It is well known that the PI3K/Akt pathway is intimately linked to the creation of an epidermal barrier, which is mostly dependent on the proliferation and differentiation of keratinocytes [137].

### 6.8. Focal Adhesion Kinase (FAK)/Src and p38 Mitogen-Activated Kinase (MAPK)

Focal Adhesion Kinase (FAK) and Src are two significant non-receptor tyrosine kinases that have been shown to play a role in the healing of wounds [138]. Initially, p38-MAPKs were referred to as the stress-activated protein kinases that are activated in cells in response to a variety of stimuli, including ultraviolet light, osmotic stress, inflammatory cytokines, changes in oxygen content, and protein synthesis inhibition [139]. Inhibitors of p38 MAPK have demonstrated anti-inflammatory properties, primarily by blocking the expression of inflammatory cytokines and controlling cellular trafficking in wounds [140]. It appears that controlling p38 MAPK activity is essential for wound healing. This apparent contradiction between normal and chronic wound healing linked to p38 MAPK shows that extracellular stress on the fibroblast potentially changes the signaling cross-talk that is linked to cell survival to increase pro-apoptotic processes [141].

### 6.9. Transforming Growth Factor/Suppressor of Mothers against Decapentaplegic (TGF-ß/Smads) and Angiopoietin-1/Tie-2 (Ang-1/Tie-2) Pathways

Transforming growth factor beta (TGF-β) signaling is required for a variety of cell functions. Although many different growth factors have been investigated as potential contributors to wound healing, TGF-β is broadly accepted to have the most far-reaching effects. TGF-β plays an important role in wound healing because of its pleiotropic effects on cell proliferation and differentiation, extracellular matrix (ECM) formation, and immunological regulation. Therapeutic drugs that target the TGF-β signaling system have shown promise in enhancing wound healing and/or diminishing scarring in preclinical investigations [142]. Five growth factors, including epidermal growth factor (EGF), platelet-derived growth factor (bFGF), vascular endothelial growth factor (VEGF), and transforming growth factor-beta 1 (TGF-1), operate as the primary regulators of cell signaling in wound healing in humans. The compounds bFGF, VEGF, PDGF, and TGF-1 stimulate cellular responses that encourage angiogenesis, keratinocyte migration and proliferation, fibroblast migration and differentiation, and collagen production. VEGF expression is increased when curcumin promotes TGF-1 signaling pathways [80]. Hesperidin is a prominent flavonoid present in lemons, sweet oranges, and a variety of other fruits, vegetables, and polyherbal mixtures. Treatment with hesperidin speeds up angiogenesis and vasculogenesis by increasing the expression of VEGF-c, Ang-1/Tie-2, TGF-β, and Smad-2/3 mRNA to help diabetic foot ulcers heal faster [79].

## 7. Clinical Trials/Human Studies on Flavonoids as a Wound Healing Agent

In this section, eight clinical studies (Table 5) that were conducted on different flavonoids for their wound-healing properties have been taken into consideration. Prior to conducting clinical trials, it is crucial to conduct in vitro and in vivo pre-clinical research. Comparing the results of pre-clinical and clinical studies will assist a researcher in bolstering the proof of research conduct. Clinical results on chronic wounds must also be examined because these results have a long-term effect on patients’ lives. Nevertheless, there are still possibilities of generating irrelevant or uncorrelated data after conducting the clinical investigations due to the mechanisms of the human body, which most of the in vitro studies are incapable of depicting.

Most flavonoid treatments for wound healing are used in conjunction with other substances that promote cell development. In a published investigation [143], the use of quercetin boosted the activity of keratinocyte proliferation. The work continued with clinical trials on 56 diabetes mellitus patients (28 men and 28 women), in which a nano-hydrogel containing quercetin and oleic acid was administered to foot skin wounds for eight months. Hyaluronic acid was used as the positive control for comparison. The combination therapy greatly outperformed the effect of the hyaluronic acid, suggesting the promising potential of quercetin as one of the flavonoid types in the management of wound healing. According to the study, quercetin effectively minimized skin lesions and improved tissue viscoelasticity and may have facilitated the treatment of chronic wounds. Modulating cytokines, growth factors, and protease were suggested to facilitate the treatment of chronic wounds while regenerating diabetic skin wounds [144]. Similarly, this capability was attributed to the exceptional antibacterial activity of quercetin. Patients also reported less discomfort and no new infections, both of which were confirmed by the trial. Neither was there any evidence of a local recurrence.

The combination of micronized purified flavonoid fraction (MPFF) with compression treatment was another flavonoid treatment combination. MPFF is a combination of diosmin and flavonoids, and this combination is expressed as hesperidin. In lieu of treating chronic venous leg ulcers with medications or natural extracts alone, the study revealed a superior technique of wound management in venous ulcers [145]. This comprises the compression treatment that uses either a bandage or a stocking in conjunction with the application of medications or other treatment, which in this instance is the MPFF. In addition, the study suggested that effective wound management may not be able to expedite recovery but has the potential to avoid recurrence. Ultrasound-guided foam sclerotherapy, endovenous laser coagulation, and radiofrequency ablation were also highlighted as treatment options in the study. The combination of compression and oral treatment with MPFF substantially hastened the healing of venous leg ulcers in 723 individuals, according to previous findings [146]. After six months of clinical testing, it was determined that MPFF can easily surpass conventional therapy by 32% in terms of healing rate. This result implies that MPFF can protect microcirculation from damage caused by elevated ambulatory venous pressure [147].

However, the treatment of wound healing with flavonoid alone is also capable of producing considerable outcomes. For instance, in a clinical experiment involving oral wounds/mucosa treated with plant-derived quercetin cream, forty male volunteers participated. The trial assessed ulcer size and pain tolerance and made general enquiries regarding the flavor and application convenience of the quercetin cream. The study revealed that the cream was capable of producing wound closure in two to four days in 35% of cases [148]. This result was found to be comparable to a study that examined anthocyanin content in a different form of adhesive gel produced from *Zea mays* and *Clitoria ternatea* extracts on oral wounds [149]. Based on the results of a clinical trial involving 68 orthodontic patients between the ages of 18 and 25, it was shown that 7 days after the application of anthocyanin gel to their dental sores, wound repair was accelerated. The clinical trial outcome was consistent with animal testing and clinical studies in which anthocyanin gel was administered to oral wounds on 60 volunteers aged 18 to 60 [150]. Quantitatively, the wound size was decreased without corresponding changes in erythema. This is related to the process of re-epithelialization, which creates new tissues. It is admissible to conclude that a flavonoid-containing mucoadhesive gel can efficiently penetrate the oral mucosa of humans and, subsequently, accelerate wound healing by stimulating fibro-blast replication [151,152].

In other instances in which flavonoid compounds were used to treat wound healing, the flavonoid type was not specified. For instance, a study was conducted on propolis, which is the common term for a substance produced by honeybees and a well-known source of flavonoids. Thirty-three patients who had uncomplicated sacrococcygeal pilonidal sinus wound surgery were evaluated for the ability of propolis to promote wound healing (from the age of 18 to 45) [153]. The wounds of these patients who had undergone marsupialization surgery were treated with a 15% propolis water solution at each dressing change, and they were examined for 28 days. In contrast to the efficacy of flavonoids in propolis based on in vitro and in vivo pre-clinical studies reported in the literature, the investigators noticed that wound-healing activity was dramatically accelerated after 14 days of therapy. However, the findings did not rule out the potential that the result was also influenced by other variables, such as the dosage given to the lesion, the source of the propolis, and the solvent-extraction technique used [154]. No local infection complications or necrosis or allergy reactions were reported in the investigation.

Another clinical trial on flavonoid-rich medicinal plants from Latin America has shed new light on the efficacy of flavonoids against persistent leg ulcers. The study was carried out with randomized, single-blind, and double-blind clinical investigations. In a single-blind study, the researcher knows about the treatment that the patients received but not vice versa. However, the double-blind method is a process in which both the researcher and the patient are uninformed of the treatments received or administered. Five medicinal plants from Latin America were compared for their wound-healing properties. Among the studied plants, *Mimosa tenuiflora* (Willd.) Poir demosntrated the strongest wound-healing activity by lowering ulcer size by 93% in the eighth week in a clinical study that involved less than 50 individuals with chronic venous leg ulcers and a one-year monitoring period. However, the study did not directly discuss the type of flavonoids [155]. It is difficult to conduct a clinical study observation on wound-healing activity in chronic venous leg ulcers because the therapy might significantly impact the patient’s life through a variety of symptoms that result in movement limitation, causing the majority of patients to discontinue treatment halfway.

It has been demonstrated that the aerial parts of *Achillea millefolium*, which is popularly known as yarrow or the common yarrow plant, contain a high level of flavonoids [156]. Extraction of the flavonoids in the plant was followed by preclinical testing that revealed wound-healing and anti-inflammatory effects [157]. As part of the continuation of a double-blind clinical investigation, 140 primiparous women served as test subjects to examine wound-healing activity following epiosomy incision. At the seventh-, tenth-, and fourteenth-day intervals, several parameters, including redness, edema, and discharge, were used to measure the healing activity. It was concluded that the treatment could lessen pain, redness, and edema, but no significant differences were observed between the treatment and control groups. Flavonoids isolated from *A. millefolium* were discovered to have no effect on wound dehiscence and secretion. In light of the possibility that the active ingredient may work in a dose-dependent manner or synergistically with other classes of secondary metabolites present in the plant extract, it is necessary to conduct additional research into the topic to understand the wound-healing property of flavonoids better. In general, the positive findings on flavonoids as wound-healing agents are somewhat restricted, and the majority of advancements did not continue until phase 3 of clinical trials to further confirm the true role of plant-based flavonoids as wound-healing agents.

## 8. Conclusions

This review provides a comprehensive assemblage of the utilization of flavonoids in the wound-healing process through diverse pathways. This serves as a foundation towards understanding and apprehending the mechanism of wound healing, thus facilitating the development of medicines to treat skin wounds.

## 9. Future Perspective and Future Study

Because flavonoid has promising potential in wound repair in regard to this study, additional natural-based extracts must be understood and integrated with cutting-edge technology with the aid of other active ingredients. For example, a flavonoid-containing chitosan hydrogel was recommended for use in the treatment of wounds due to its positive effect on wound-healing induction and antioxidant activity in diabetic mice [158]. The enhanced production of anionic phenolic hydroxyl groups in chitosan fibers has been shown to greatly aid antioxidant and wound-healing activities [159]. Chitosan is extensively used as a medicine carrier due to its antimicrobial, biocompatible, biodegradable, and non-toxic properties [160]. In particular, a drug delivery system that utilizes natural-based treatments for wound healing can be enhanced by adding nanotechnology. Bio-nanomaterials derived from natural sources may be one of the most promising means of accelerating tissue repair. In a recent study on the efficacy of flavonoid, which was efficiently induced in bio-fabricated nano-biomaterials, flavonoid-loaded silver derived from the seed of the *Madhuca longifolia* plant was found to be effective [161]. Compared to flavonoid-loaded gold and bimetallic substances, flavonoid-loaded silver enhanced wound healing by up to 80.33%. In another recent study, chronic wounds treated with carbonized nanogel (copper sulfide nanoclusters) and quercetin exhibited fast healing of wounds. The discovered multifunctional nanogel can stimulate angiogenesis, epithelialization, and collagen synthesis to accelerate granulation tissue formation [162].

On the contrary, according to the literature, the majority of studies have not yet conducted in vivo testing or clinical trials. Similarly, not all conducted clinical studies have yielded consistent findings, and the majority of clinical trials did not even reach Phase 3. By focusing on a single primary source of flavonoids, more therapeutic investigations can be conducted. Likewise, more research confirming flavonoids’ efficacy in all therapeutic domains is recommended. The current study has the potential to introduce a novel concept to flavonoid discovery, highlighting the signaling pathways for future endeavors. This work provides future researchers with additional information on the limitations of flavonoid therapy for tissue regeneration. Thereby, the results of this study can serve as a starting point for future research to select the optimal signaling pathways that will result in the quickest rate of healing without scarring and will not activate unwanted (malignant) or negative pathways.

## Figures and Tables

**Figure 1 ijms-24-04607-f001:**
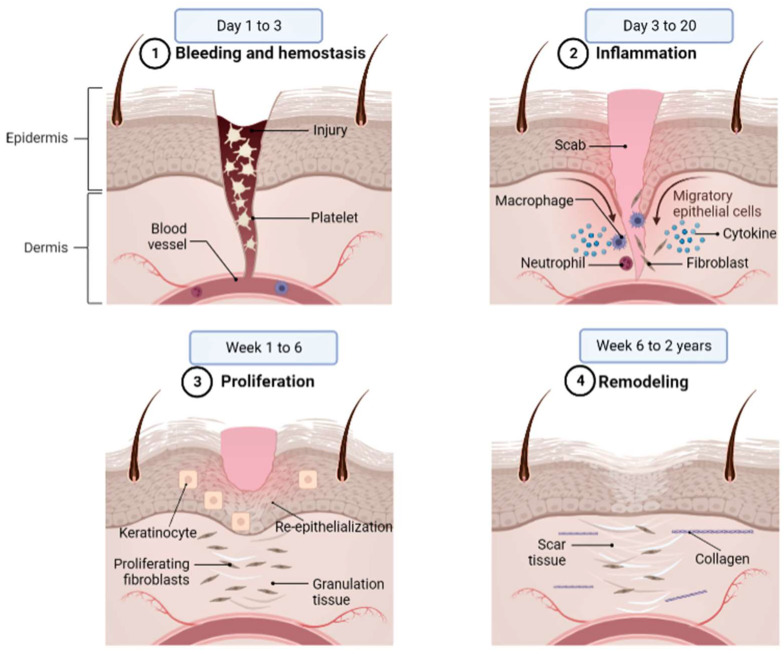
Physiological wound-healing phases. ➀ Bleeding and hemostasis: hemostasis occurs immediately after bleeding and reduces blood flow by constricting blood vessels to stop bleeding following vascular injury. ➁ Inflammation: the release of cytokines, growth factors, and the migration of leukocytes into the area cause hemostasis and acute inflammation in the area of a wound. ➂ Proliferation: an increase in the keratinocyte, fibroblast, endothelial, and leukocyte migration and proliferation in the wound area. ➃ Remodeling: the process in which stretching and pulling forces interact with the scar tissue. They help collagen to regain the normal alignment that is needed.

**Figure 2 ijms-24-04607-f002:**
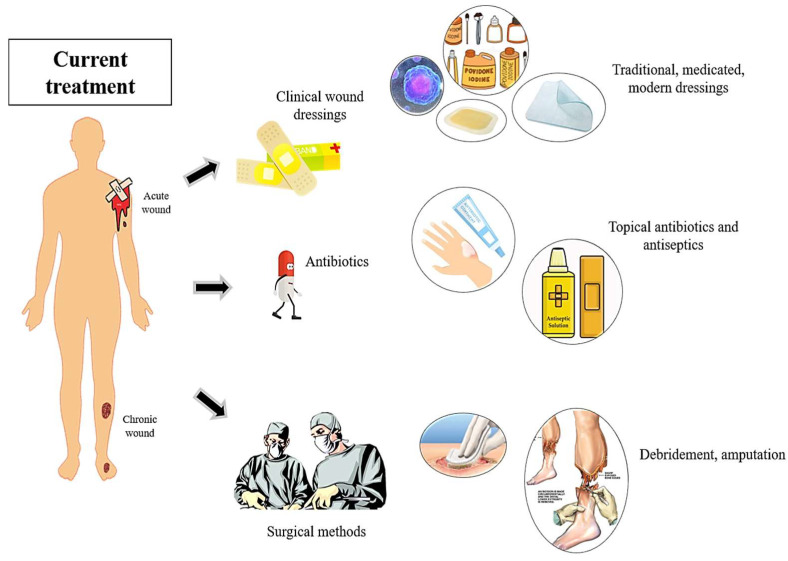
Current available treatments for wound healing for acute and chronic wounds. The methods include traditional and modern approaches as well as the treatment technique.

**Figure 3 ijms-24-04607-f003:**
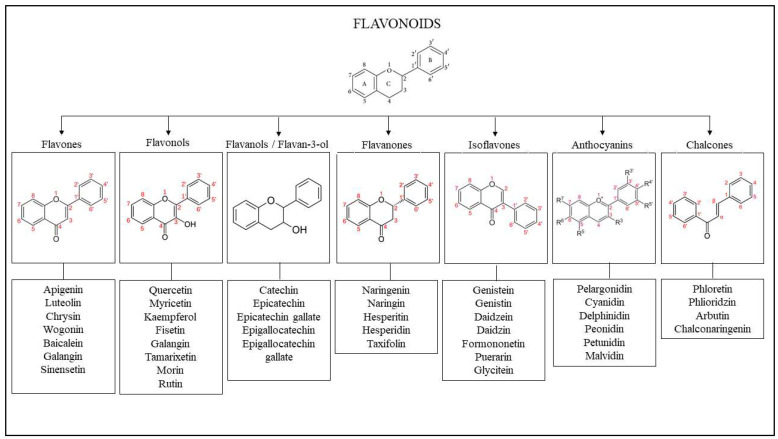
Chemical structures of flavonoids and their major classes along with the examples.

**Figure 4 ijms-24-04607-f004:**
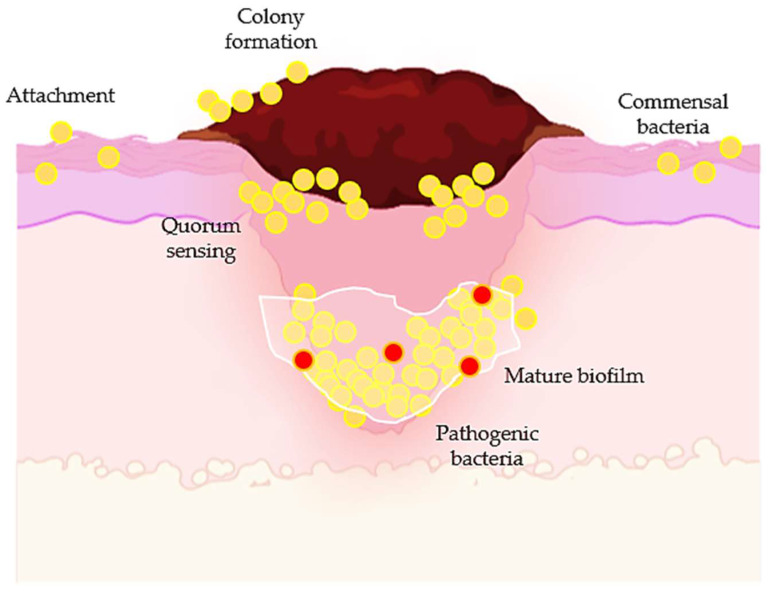
The colonization of commensal bacteria as the injury occurs. Biofilm production may cause delayed wound healing and treatment failures, as well as susceptibility to new infections, as the production of biofilms prevents drug penetration.

**Figure 5 ijms-24-04607-f005:**
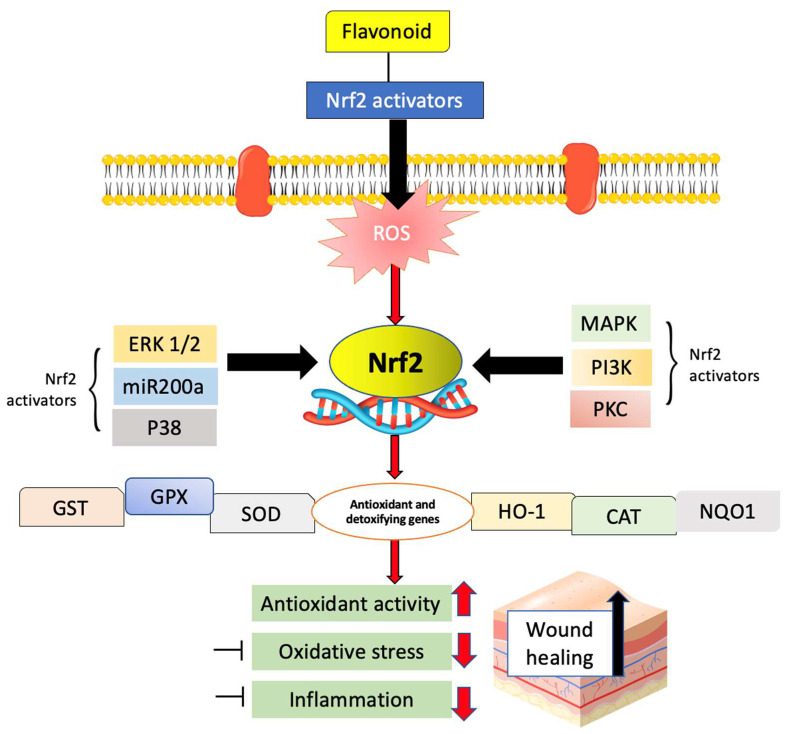
Potential mechanisms that Nrf2 activation affects wound healing. Flavonoids act as the Nrf2 activators that activate the ERK1/2, p38MAPK, PI3K, and PKC signaling pathways. Extracellular signal-regulated protein kinases 1 and 2 (ERK1/2), mitogen-activated protein kinases (MAPK), phosphoinositide 3-kinase (PI3K) and protein kinase C (PKC), glutathione S-transferase (GST), glutathione peroxidase 1 (GPX), superoxide dismutase type (SOD), heme oxygenase-1 (HO-1), catalase (CAT), and NAD(P)H quinone dehydrogenase 1 (NQO1).

**Table 1 ijms-24-04607-t001:** The different types of dressings currently used to treat chronic wounds.

Type of Dressings	Available Treatments	Functions	Limitations	Refs.
Medicated	Growth factor	Promotes tissue regeneration by angiogenesis and cellular proliferation.	The suitability for appropriate dressings is challenging for effective release at the wound site.	[40,41,42]
	Antimicrobials	Prevents infections.	Cell and organ toxicity due to a high dose of prescription antibiotic.	[43]
Traditional	Topical applications, e.g., povidone iodine, saline	Cleanse wound and irrigates dry wound.	Applicable only for small wounds and suppuration.	[44]
Modern	Hydrocolloids	Prevents mild exuding wounds from drying out and offers moist insulation for wound healing to take place.	Fibers are deposited upon dressing and need to be replaced during dressing change.	[45]
	Foam	Provides prolonged moisture and effective absorbent.	Not applicable to dry scars and epithelial wounds.	[46]
	Bioactive, e.g., collagen, hydrogel	Promotes matrix formation and is biodegradable.	Required additional testing to compare with standard wound dressing.	[47,48]

**Table 2 ijms-24-04607-t002:** Summary of flavonoids efficacy in in vitro and in vivo studies.

Years	Flavonoid Types	Study Design	Efficacy of Flavonoids Treatment	Refs.
2019	Quercetin-3-oleate	In vitro study on HaCaT cell.	Quercetin stimulated the HaCaT cell by 51% compared to the untreated cell. The result suggested an increment in TGF-β production and MMP-9 release.	[77]
2018	Hesperidin	In vivo study on Sprague–Dawley rats.	Hesperidin enhanced healing in the chronic diabetic wound by up-regulating VEGF-c, Ang-1/Tie-2, TGF-β, and Smad-2/3 mRNA expression.	[79]
2021	Rutin	In vivo study on mice model.	Reduction of wound area was observed over a period between the 4th and 16th days. H2 hydrogel formulation was chosen as the best concentration to accommodate wound-healing activity.	[83]
2020	Morin (flavonoid extracted from *Maclura pomifera* plant)	In vivo study on 44-week-old mice.	Morin stimulated the Wnt pathway during osteoblasts development and promoted angiogenesis, thus, portraying a promising future in bone defects regeneration treatment.	[84]
2018	Quercetin	In vivo study on albino rats.	The quercetin loaded liposomal hydrogel accelerated fibroblast cell growth in 4 days of treatment on subjects with shaved back wound.	[72]
2019	Kaempferol	In vivo study on diabetic and non-diabetic rats.	Kaempferol was suggested to be an effective wound-healing agent because it was able to induce a 92.12% healing rate.	[87]

**Table 3 ijms-24-04607-t003:** The application of flavonoids as anti-bacterial agents on wound healing.

Samples	Models	Parameters	Results	Ref.
*Juglans regia* leaves	*Staphylococcus lugdunensis*, *Proteus vulgaris*, and *Staphylococcus epidermidis.* HaCaT cells.	Minimal inhibitory concentrations (MIC) and minimal bactericidal concentrations (MBC). Wound scratch healing. Cell invasion.	HPLC profiles revealed 10 flavonoids and its derivatives. MIC of 2–4 mg/mL against *S. lugdunensis* and *S. aureus.* Good inhibitory effects against Gram-negative strains (*P. vulgaris* and *S. lugdunensis*) as compared to Gram-positive strains (*S. epidermidis* and *S. aureus*). About 28% of wound closure was recorded 40% inhibition against *P. vulgaris-*infected HaCaT cells.	[93]
Quercetin and 4-formyl phenyl boronic acid (4FPBA-Q complex)	Multidrug resistance of *Salmonella typhi*, *S. aureus*, *Pseudomonas aeruginosa.* Male Wistar albino rats.	MIC and MBC. Primary dermal irritation index. In vivo wound healing.	The 4FPBA-Q complex was more effective against *P. aeruginosa* and *S. typhi.* No sign of erythema upon 4FPBA-Q complex. Treatment with 4FPBA-Q complex demonstrated ample healing in 10 days.	[94]
Quercetin and Curcumin	*S. aureus* and *P. aeruginosa.* Human dermal fibroblasts (HDFB).	Disc diffusion assay. In vitro scratch assay. Migration assay.	A 1:1 ratio of quercetin/curcuminoid exhibited the strongest inhibition zone against both strains. Single compound did not show any inhibition. The percentage of wound closure is higher upon treatment with quercetin alone or in combination with curcuminoid. A 3:1 ratio of quercetin/curcuminoid significantly induced HDFB migration toward matrix crossing.	[90]
*Parkia clappertoniana* fruit husk extract (PCFHE)	Male Sprague–Dawley rats. *S. aureus*, *Bacillus subtilis*, *Escherichia coli*, *P. aeruginosa*, *Klebsiella pneumoniae.*	Excision wound model. H&E and collagen staining. Skin irritation test. MIC and MBC.	PCFHE-treated group showed potential wound-healing properties during epithelization and increased collagen content. PCFHE showed concentration dependent inhibition on all microbial growth except *S. aureus.*	[95]
*Coccinia grandis* leaves	*Bacillus cereus.* Male Wistar albino rats.	Daily observation of wound closure. Histopathological examination.	LC-ESI-MS/MS displayed that the extract was rich in flavonoids. Good healing ability and almost similar to the Fucidin-treated group. H&E staining portrayed complete re-epithelization of epidermis. Deposition of collagen fiber was as good as Fucidin-treated group.	[96]
*Lafoensia capari* leaves (HELp)	*S. aureus*, *S. epidermidis*, *Streptococcus pyogenes*, *Enterococcus faecalis*, *S. typi*, *P. aeruginosa*, *Shigella flexneri*, *K. pneumoniae*, *E. coli.* RAW 264.7 murine macrophages, Chinese hamster ovary epithelial cells, and L929 fibroblasts. Albino mice (*Mus musculus*), Swiss-Webster strain, and rats (*Rattus norvegicus*), Wistar strain.	Excision wound-healing model. Scratch wound-healing assay. Western blot analysis of *p*-ERK1/2 in vitro.	The contraction rates increase with a topical application of HELp on Day 2. At 10 and 30 mg/gel, HELp exhibited moderate re-epithelization and neovascularization through H&E staining. The rate of fibroblast migration increases by 25.1% and 35.3% at 0.1 and 0.03 ug/mL, respectively. HELp upregulated the expression of *p*-ERK1/2, which promoted cell proliferation. ESI-MS revealed that HELp was rich in phytochemicals, i.e., flavonoids that contributed to the wound-healing properties for in vivo and in vitro findings.	[97]

**Table 4 ijms-24-04607-t004:** Wound healing effects of flavonoids according to respective pathways.

Pathways	Flavonoids	Experiments	Outcomes	Gene/Protein Detection	Refs.
Wnt/β-catenin	Quercetin	In vitro: CCK-8 and scratch assay (skin cells) In vivo: Histological staining (C57BL/6 mice), Western blot, RT-qPCR analysis, and molecular docking	Wound healing rates increased in dose-dependent manner compared to the control group; levels of inflammatory factors, including tumor necrosis factor-alpha, interleukin-1 beta and interleukin-6 were significantly reduced after quercetin administration; improved level of GSH; molecular docking analysis validated the formed hydrogen bonds between quercetin and Ala195, Gln308, Asn369, and Lys372 residues of TERT.	Telomerase reverse transcriptase (TERT)	[98]
	Flavonoids with two OH groups in the B-ring, such as sterubin, luteolin, and hydroxygekwanin	In vivo: Dorsal skin samples (2.25 cm) were excised from 7-week-old C57BL/6 mice; flavonoids (0.1% *w*/*v*) were dissolved in 50% ethanol; and 200 µL of the flavonoid solution was applied daily to the wounded area for 1 week.	Flavonoids with two OH groups in the B-ring, such as sterubin, luteolin, and hydroxygenkwanin, showed promising effects in regenerating black pigmented hairs, whereas those with one OH group in the B-ring showed no significant change.	N/A	[99]
TGF-β	Hesperidin	In vivo: Diabetes was induced experimentally by streptozotocin (STZ, 55 mg/kg, i.p.) in Sprague–Dawley rats (180–220 g), and hesperidin (25, 50 and 100 mg/kg, p.o.) was administered for 21 days after wound stabilization. Various biochemical, molecular, and histopathological parameters were evaluated in wound tissue.	Hesperidin treatment showed a significant increase (*p* < 0.05) in percent wound closure and serum insulin level. Intraperitoneal administration of STZ caused significant down-regulation in VEGF-c, Ang-1, Tie-2, TGF-beta, and Smad 2/3 mRNA expression in wound tissues, whereas hesperidin (50 and 100 mg/kg) treatment showed significant up-regulation in these mRNA expressions.	VEGF-c, Ang-1/Tie-2, TGF-β and Smad-2/3 mRNA	[79]
Notch	Vaccarin	In vitro: In this study, the EAhy926 cells were exposed to 250, 500, and 1000 M H_2_O_2_ for 2 and 4 h in the absence or presence of vaccarin, and the cell injury induced by H_2_O_2_ was examined via sulforhodamine B (SRB) assay. Cell migratory ability, lactate dehydrogenase (LDH) leakage, malondialdehyde (MDA) levels, and decreasing superoxide dismutase (SOD) activity were evaluated by the wound-healing assay and corresponding assay kits.	Western blot detected the protein expressions of Notch1, Hes1, and caspase-3. Preincubation with vaccarin was found to protect EA.hy926 cells from H_2_O_2_-induced cell oxidative stress injury, which promoted cell viability and cell migratory ability and inhibited the level of LDH and MDA but enhanced the activity of SOD. In addition to the downregulation of Notch signaling, vaccarin treatments also downregulated caspase-3, a cell-apoptotic-pathway-related protein.	Notch1, Hes1 and Caspase-3	[100]
Bone morphogenetic protein (BMP) pathway	Icariin	In vitro: CCK-8 assay and live/dead cell staining, immunofluorescence staining, Western blotting, RT-PCR, histological analysis, and cell migration assay	Icariin+PEG hydrogel resulted in faster healing and formation of new hair follicles; induced a higher level of M2 phenotypic transformation of macrophages; reduced the invasion of inflammation, excessive deposition of collagen, and immoderate activation of myofibroblasts; and increased the expression of BMP4 and Smad1/5 phosphorylation in skin wounds.	BMP4; Smad1/5	[101]
	Quercetin	In vitro: RNA analysis In vivo: Wild-type C57BL/6J mice were treated with quercetin (0.5, 1, 5, or 50 mM). Corneal scarring was assessed for 3 weeks by slit lamp imaging and clinically scored. In a separate animal study, six New Zealand White rabbits underwent lamellar keratectomy surgery, followed by treatment with 5 mM quercetin or vehicle twice daily for three days. Stromal backscattering was assessed at week 3 by in vivo confocal microscopy.	In vitro: Human corneal fibroblasts showed that quercetin modulated select factors of the transforming growth factor-beta (TGF-beta) signaling pathway. These results provide evidence that quercetin may inhibit corneal scarring. In vivo: In mice, a single dose of 5 mM quercetin reduced corneal scar formation. In rabbits, stromal backscattering was lower in two out of three animals in the quercetin-treated group.	TGF-β2 and SMAD7	[102]
Hedgehog, bone morphogenetic protein (BMP), and Wnt signaling	Propolis	In vivo: GC-MS analysis and antioxidant activity testing; blood glucose levels testing; real-time polymerase chain reaction method.	Phenols and flavonoids from the ethanolic extract of propolis (EEP) can improve the caudal fin regeneration of hyperglycemic zebrafish.	Shha, Igf2a, Bmp2b, and Col1a2	[103]
PI3K/AKT/mTOR	Isoliquiritigenin (ISL)	In vitro: MTT assay, wound-healing assay, quantitative real-time PCR, western blot and immunofluorescence assay.	The results showed that ISL inhibited TGF-beta 1-induced proliferation and migration and down-regulated the expressions of alpha-smooth muscle actin (α-SMA), collagen type I alpha 1 (COLIA1), and fibronectin (FN). ISL treatment led to up-regulation of microtubule-associated protein light chain 3 (LC3) in TGF-beta 1-treated MRC-5 cells, accompanied by a significant decrease in the phosphorylation levels of phosphatidylinositol 3-kinase (PI3K), protein kinase B (AKT), and mammalian target of rapamycin (mTOR).	α-SMA, COLIA1 and FN	[104]
Mitogen-activated protein (MAP) kinases, such as ERK1,2 and Jun N-terminal kinase (JNK) 1,2 activation and c-Jun phosphorylation	Oleanolic acid (OA)—pentacyclic triterpene	In vitro*:* Scratch assay in two epithelial cell lines of different linage: non-malignant mink lung epithelial cells, Mv1Lu, and human breast cancer cells, MDA-MB-231.	OA enhanced cell migration for in vitro scratch closure; MDA-MB-231 cells treated with OA displayed an altered gene expression profile affecting transcription factor genes (*c-JUN*) as well as proteins involved in migration and ECM dynamics (*PAI1*); OA treatment served changes in the epidermal growth factor receptor (EGFR) subcellular localization.	phospho-c-Jun, c-Jun, phospho-ERK1/2 and phospho-JNK1/2 proteins	[105]
Notch, AKT, and MAPK signaling pathways.	Total flavonoids from *Semen Cuscutae* (TFSC), the main estrogenic active constituent of the plant, which includes hyperin, rutin, quercitrin, quercetin, and isorhamnetin	In vitro: HTR-8 cells migration and invasion functions were analyzed using wound healing and transwell assays. The regulatory effect of TFSC on MMP9 expression and relevant signaling. pathways were analyzed by western blot.	TFSC significantly promoted the migration of EVT cells in a dose and time-dependent manner compared to control group. The migration and invasion of EVT cells were maximized at the highest dosage of 5 μg/mL of TFSC. The expression of MMP9 in EVT cells was significantly increased after TFSC treatment. Furthermore, cells treated with TFSC significantly upregulated protein expressions in Notch, AKT, and p38/MAPK signaling pathways.	MMP9	[106]
EGFR/PI3K/AKT	Chinese medicine Huiyang Shengji decoction (HYSJD) main components were flavonoids, terpenes, alkaloids, phenylpropanoids, and carbohydrates	In vivo: Microarray analysis, GO and KEGG enrichment analysis, ELISA assays, and western blot analysis	HYSJD was found to increase the wound-healing rate in chronic skin ulcers in mice at days 3, 7, and 14 post-wound formation and promote the proliferation of epidermal cells. Two proteins that were differentially expressed between the different groups, i.e., IGF-1 and EGFR, were further validated. Serum ELISA assays showed that serum EGFR in the HYSJD treatment group was significantly increased. KEGG pathway analysis suggested that the PI3K/AKT pathway involved in HYSJD promoting the proliferation of epidermal cells in chronic wounds in mice.	IGF-1 and EGFR	[107]
MAPK and PI3K/Akt	Pinocembrin	Cell viability assay; direct measurement of cell number; cellular wound-healing activity assay; western blot analysis; immunofluorescence study	Pinocembrin induced an increase in HaCaT cell number and significantly triggered ERK1/2 and Akt activation. Pinocembrin induces keratinocyte proliferation mainly by activating MAPK and PI3K/Akt kinases.	pAKT pERK	[108]
Akt and ERK	Icariin	In vitro: CCK-8 assay; transwell assay; western blot assay; RT-qPCR; ELISA. In vivo: hematoxylin and eosin (H&E) staining; epidermal thickness assessment.	In vitro shows that icariin significantly promoted the migration and proliferation of keratinocytes via the activation of AKT serine/threonine kinase 1 (AKT) and extracellular signal-regulated kinase (ERK). In addition, icariin inhibited the production of interleukin (IL)-6 and tumor necrosis factor (TNF)-α and induced the production of IL-10. In vivo shows icariin treatment accelerated the wound closure rate.	Cyclin D1 and D3	[109]
ERK, P38, JNK and Akt	Artocarpin	In vitro: Cell viability; proliferation assays; wound healing assay. In vivo: BrdU incorporation assay; mouse cytokine array; immunohistochemical staining; immunoblotting analysis; Matrigel assay. *Excisional wound model; Transmission electron microscopy (TEM).*	Artocarpin accelerated inflammatory progression and subsequently decreased persistent inflammation. Artocarpin increased collagen production and increased human fibroblast proliferation and migration by activating the P38 and JNK pathways. Moreover, Artocarpin increased the proliferation and migration of human keratinocytes through the ERK and P38 pathways and augmented human endothelial cell proliferation and tube formation through the Akt and P38 pathways.	Akt, ERK, and P38	[110]

**Table 5 ijms-24-04607-t005:** Summary of flavonoids on wound-healing clinical trials.

Year	Flavonoids/Plants	Clinical Trial Design	Observation Time (Day)	Results	Refs.
2020	Quercetin (incorporated with oleic acid)	56 patients (28 for both men and women) who applied the nano-hydrogel containing quercetin and oleic acid on the wound.	8 months	Quercetin reduced skin lesion, improved tissue viscoelasticity, and could facilitate treatment in chronic wound.	[143]
2009	Micronized purified flavonoid fraction (MPFF) + compression treatment	723 patients who took the MPFF orally.	6 months	The combination of MPFF with the compression method sped up recovery of venous leg ulcer.	[145]
2010	Quercetin	40 male volunteers were given cream containing quercetin to apply on their oral ulcer.	10 days	Quercetin cream is able to relieve pain and induced complete healing within 2–4 days.	[148]
2018	Anthocyanin (water-soluble flavonoid)	68 orthodontic patients were given anthocyanin gel to apply on their oral wound.	7 days	The mucoadhesive gel containing anthocyanin promoted wound closure in acute oral wounds	[149]
2021	Anthocyanin	A double-blinded clinical trial on 60 volunteers was conducted with the application of anthocyanin gel onto oral wounds.	7 days	The anthocyanin niosome gel accelerated wound closure, thus improving patients’ life quality in terms of relieving pain from the oral wound.	[150]
2021	Flavonoid class is not mentioned specifically	33 patients for uncomplicated sacrococcygeal pilonidal sinus wound surgery.	28 days	The treatment of 15% propolis water solution facilitated wound recovery by showing significant healing rate starting from the first week.	[153]
2022	Flavonoid class is not mentioned specifically	The study categorized the clinical trials into three groups: randomized, single-blinded, and double-blinded. 50 subjects for chronic venous leg ulcers	8 weeks	The study analyzed 305 plant species with wound-healing properties. Among the plants, 25 compounds that contained substantial levels of flavonoids were isolated. *Mimosa tenuiflora* (Willd.) poir extract exerted the highest wound-healing activity by reducing the ulcer size to 93% in 8th weeks.	[155]
2007	Flavonoid class is not mentioned specifically	A double-blind clinical trial was continued by having 140 primiparous women as test subjects to study wound-healing activity on epiosomy incision	14 days	The treatment can reduce pain, redness, and edema. No significant difference was found for both treatment and control groups in reducing the likelihood of wound dehiscence and secretion. The study suggested the importance of using appropriate dosage in the treatment.	[156]

## Data Availability

Not applicable.

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
