# Peer review of "Flavonoids as Potential Wound-Healing Molecules: Emphasis on Pathways Perspective"

_ijms, 2023, doi:10.3390/ijms24054607_

Round 1
Reviewer 1 Report
The paper can be published in IJMS after the minor revision. The subject of the paper corresponds to the scope of special issue. The authors did a literature review on the use of flavonoids in wound healing. The article is a review based on the latest literature on the subject. The authors summarize the knowledge on this subject in a practical way and set trends for the future. I just have a few comments that can improve the work:
- Line 292 “Dietary…vegetables” - this phrase is redundant
- Line 299 “Figure 3” why bold?
- Line 351 “rutin flavonoid glucoside”? should be glycoside
- Line 359 mg/kg/d? “d” means day?
- Line 378 “by” – is unnecessary
- Table 2 - There is no reference to the table in the text. The table headers have probably been copied from table 1 and do not match the contents of table 2.
- Line 449 - no literature reference
- Lines 445-595 - there are no valid literature references, only author and year
- Table 3 - There is no reference to the table in the text. The heading "flavonoids" does not match the content, for example propolis is not a flavonoid.
- Line 644 “study by [104]” and line 650 “conducted by [105]” - stylistic error
- Line 686 “[111]”, and line 688 “[112]” - why are the references deleted?
- Line 693 should be “millefolium”
- Table 4 There is no reference to the table in the text. Oleic acid is neither a flavonoid nor a plant. What is MPFF?
- Line 726 should be “longifolia”
- Line 908 double “79”
Author Response
Manuscript ID: ijms-2143070
Manuscript Title: Flavonoids as Potential Wound Healing Molecules: Emphasis on Pathways
Perspective
REVIEWER 1:
The paper can be published in IJMS after the minor revision. The subject of the paper corresponds to the scope of special issue. The authors did a literature review on the use of flavonoids in wound healing. The article is a review based on the latest literature on the subject. The authors summarize the knowledge on this subject in a practical way and set trends for the future. I just have a few comments that can improve the work:
Response To Reviewer:
We appreciate and wish to thank you for the useful reviewers’ comments, suggestions, and the opportunity to improve the manuscript. We have revised the manuscript, including those based on the comments/suggestions accordingly, and the detailed corrections are listed below in the authors’ response column. The changes in the manuscript are highlighted in yellow.
|
No. |
Reviewers’ comments |
Authors’ response |
|
1) |
Line 292 “Dietary…vegetables” - this phrase is redundant |
Line 347 - The word dietary has been removed. “Flavonoids are abundant in fruits and vegetables. Flavonoids are phenolic compounds that can be found in fruits, vegetables, herbs, cocoa, chocolate, tea, soy, red wine, and other plant food and beverage products.” |
|
2) |
Line 299 “Figure 3” why bold? |
Line 354 - The bold format has been removed |
|
3) |
Line 351 “rutin flavonoid glucoside”? should be glycoside |
Line 358 – Corrected. Rutin is already a glycoside. Just mention it as rutin only. |
|
4) |
Line 359 mg/kg/d? “d” means day? |
Line 435 – 100 mg/kg/d changed to 100 mg/kg per day |
|
5) |
Line 378 “by” – is unnecessary |
Line 454 – deleted “by” and replaced with new rephrasing. |
|
6) |
Table 2 - There is no reference to the table in the text. The table headers have probably been copied from table 1 and do not match the contents of table 2. |
Line 406 – Table 2 has been referred to in the text. “Table 2 summarizes the effect of flavonoids treatment in in vivo and in vitro testing.” Line 476 - Table 2 headers had been matched to its respective contents. |
|
7) |
Line 449 - no literature reference Lines 445-595 - there are no valid literature references, only author and year. |
Line 637 – 787 - All valid literature references have been included. |
|
8) |
Table 3 - There is no reference to the table in the text. The heading "flavonoids" does not match the content, for example propolis is not a flavonoid. |
Line 790 – Table 3 has been renamed to Table 4. Table 4 has been included in the text. Propolis under “flavonoids” heading changed to “flavonoids from propolis” |
|
9) |
Line 644 “study by [104]” and line 650 “conducted by [105]” - stylistic error |
Line 839 & 844 – Stylistic error had been corrected |
|
10) |
Line 686 “[111]”, and line 688 “[112]” - why are the references deleted? |
Line 880 & 882 - Rreferences included |
|
11) |
Line 693 should be “millefolium” |
Line 888 - A. milleforium changed to A. millefolium |
|
12) |
Table 4 There is no reference to the table in the text. Oleic acid is neither a flavonoid nor a plant. What is MPFF? |
Line 896 – Table 4 has been renamed to Table 5. Table 5 has been referred to in the text. Rephrasing is also done – “Quercetin (incorporated with oleic acid)” MPFF = Micronized purified flavonoid fraction |
|
13) |
Line 726 should be “longifolia” |
Line 1020 - Madhuca longifilia changed to Madhuca longifolia |
|
14) |
Line 908 double “79” |
Line 1284 - Double 79 deleted |

Reviewer 2 Report
Prezados autores, parabenizo pelo trabalho que reúne informações sobre um composto fenólico particularmente responsável por diversas atividades biológicas. Nesse sentido, o papel dos flavonoides no processo de cicatrização ou tratamento de feridas foi o foco principal desta pesquisa de revisão. No entanto, para ser aceito para publicação, aqui estão minhas considerações:
1. INTRODUÇÃO
- Nicotine and cocaine were used as healing agents until the beginning of the 19th century, from 1914 cocaine was banned in the USA due to its psychoactive effects. In addition, nicotine was classified as a potential cause of chemical dependence. I suggest that when mentioning these substances, the historical factor and the main reasons for not using them "currently" for the healing process are described.
- Regarding the 3 types of chronic wounds, it is important to briefly report the characteristics that differentiate diabetic ulcers from pressure or vascular ulcers.
- In lines 145 and 146 the term "wounds" is repeated 3 times in two lines, I suggest that in order not to be so repetitive, the term be changed to "injury" when convenient.
- In line 172 the authors report that the treatment of acute and chronic wounds is the most difficult problem in the world. I suggest reviewing this information.
- Line 178. I suggest that flavonoids be better described in this line. " Flavonoids are secondary metabolites present in several types of vegetables, being frequently reported in the literature as responsible for numerous biological activities, including the treatment of wounds. ".....
6. Clinical Trials/Human Studies on Flavonoids as a Wound Healing Agent
- In this item it was clear the change in the writing used in the manuscript, use of colloquial language. In addition, the item is confusing and without connection between the paragraphs. I suggest reviewing this entire item.
8. Future Perspective and Future Study
- I suggest a general review of this item addressing what are the prospects for future studies that this work may contribute.
Conclusion of the review:
- Dear authors, I suggest a thorough revision of the organization of the manuscript. Several times it was possible to observe information about flavonoids in different paragraphs, making the text tiring, confusing and not very explanatory. In this sense, I suggest that the text be organized so that there is an objective sequence of information. Ex:
1. Wounds and their morphological presentations
2. Conventional treatments
3. Socioeconomic effects of morbidity and mortality from chronic wounds
4. Inclusion of natural products as an alternative for treating wounds
5. Flavonoids
6. Mechanisms of action of flavonoids in wound repair
Com o texto bem definido apresentando uma sequência objetiva que ajude o leitor a entender o estudo, o manuscrito será indicado para publicação.
Author Response
Manuscript ID: ijms-2143070
Manuscript Title: Flavonoids as Potential Wound Healing Molecules: Emphasis on Pathways
Perspective
REVIEWER 2:
Prezados autores, parabenizo pelo trabalho que reúne informações sobre um composto fenólico particularmente responsável por diversas atividades biológicas. Nesse sentido, o papel dos flavonoides no processo de cicatrização ou tratamento de feridas foi o foco principal desta pesquisa de revisão. No entanto, para ser aceito para publicação, aqui estão minhas considerações:
Dear authors, I congratulate you for the work that gathers information about a phenolic compound particularly responsible for several biological activities. In this sense, the role of flavonoids in the healing process or wound treatment was the main focus of this review research. However, to be accepted for publication, here are my considerations:
Com o texto bem definido apresentando uma sequência objetiva que ajude o leitor a entender o estudo, o manuscrito será indicado para publicação.
With the well-defined text presenting an objective sequence that helps the reader to understand the study, the manuscript will be indicated for publication.
Response To Reviewer:
We appreciate and wish to thank you for the useful reviewers’ comments, suggestions, and the opportunity to improve the manuscript. We have revised the manuscript, including those based on the comments/suggestions accordingly, and the detailed corrections are listed below in the authors’ response column. The changes in the manuscript are highlighted in yellow.
|
No. |
Reviewers’ comments |
Authors’ response |
|
|
1. Introduction |
|
|
1) |
Nicotine and cocaine were used as healing agents until the beginning of the 19th century, from 1914 cocaine was banned in the USA due to its psychoactive effects. In addition, nicotine was classified as a potential cause of chemical dependence. I suggest that when mentioning these substances, the historical factor and the main reasons for not using them "currently" for the healing process are described. |
Line 91 – The historical factors and the main reasons for not using them "currently" for the healing process have been included.
“In the early 19th century, nicotine and cocaine were both used medicinally but cocaine was outlawed in the US in 1914 due to its hallucinogenic properties. Additionally, nicotine was highlighted as a potential contributor to pharmaceutical overreliance [10, 11]. Meanwhile back in 1808, ergotamine was widely used to precipitate childbirth and to control post-partum hemorrhage due to its remarkable uterotonic and vasoconstrictor effects [12]. European Medicines Agency's Committee for Medicinal Products for Human Use (CHMP) has suggested prohibiting the use of medications that include ergot derivatives. Since the hazards outweigh the benefits in these indications, these medications should no longer be used to treat a variety of illnesses involving blood circulation issues [13].”
|
|
2) |
Regarding the 3 types of chronic wounds, it is important to briefly report the characteristics that differentiate diabetic ulcers from pressure or vascular ulcers |
Line 57 – Characteristics of the differences between diabetic ulcers, pressure ulcers, and vascular ulcers have been briefly reported. “It is understood that pressure ulcers can develop over pressure points, such as the heel of a bedridden patient or the side of the foot from wearing tight shoes. Conventionally, any ulcer in a diabetic patient is interpreted to as a diabetic ulcer. While vascular reflux is most likely the cause of ulcers that develop at the ankle, calf, or pretibial sites [5].” |
|
3) |
In lines 145 and 146 the term "wounds" is repeated 3 times in two lines, I suggest that in order not to be so repetitive, the term be changed to "injury" when convenient. |
Line 163 – Repetitive term “wounds” have been substituted for injury and rephrasing has been done accordingly. “The wound physically shrinks during the healing process which is thought to be mediated by contractile fibroblasts (myofibroblasts) that develop in the injury [19, 21].” |
|
4) |
In line 172 the authors report that the treatment of acute and chronic wounds is the most difficult problem in the world. I suggest reviewing this information. |
Line 189 – Has been reviewed accordingly and corrected. “Regrettably, the management of both acute and chronic wounds continues to be one of the most challenging healthcare issues in the world.” |
|
5) |
Line 178. I suggest that flavonoids be better described in this line. " Flavonoids are secondary metabolites present in several types of vegetables, being frequently reported in the literature as responsible for numerous biological activities, including the treatment of wounds. "..... |
Line 197 – Corrected as per the suggestion. Thanks "Flavonoids are secondary metabolites present in several types of vegetables, being frequently reported in the literature as responsible for numerous biological activities, including the treatment of wounds [15].”
|
|
|
Subtopic 6. Clinical Trials/Human Studies on Flavonoids as a Wound Healing Agent |
|
|
6) |
In this item it was clear the change in the writing used in the manuscript, use of colloquial language (informal language). In addition, the item is confusing and without connection between the paragraphs. I suggest reviewing this entire item.
|
Line 792 – Has been thoroughly reviewed as per the constructive suggestion. Moreover, subtopic 6 has been renamed to subtopic 7 due to the reorganization of this manuscript.
|
|
|
Subtopic 8. Future Perspective and Future Study |
|
|
7) |
I suggest a general review of this item addressing what are the prospects for future studies that this work may contribute.
|
Line 1007 – Contributions of this work for future studies prospects have been addressed. “Since flavonoid has promising potential in wound repair in regard to this study, additional natural-based extracts must be comprehended and integrated with cutting-edge technology with the aid of other active ingredients. For example, chitosan hydrogel containing flavonoid was recommended for use in the treatment of wounds due to its positive effect on wound healing induction and antioxidant activity in diabetic mice [165]. The enhanced production of anionic phenolic hydroxyl groups in chitosan fibers has been shown to greatly aid antioxidant and wound healing activities [166]. Chitosan is extensively used as a medicine carrier due to its antimicrobial, biocompatible, biodegradable, and non-toxic properties [167]. In particular, the drug delivery system utilizing natural-based treatments for wound healing can be enhanced by adding nanotechnology. Bio-nanomaterials derived from natural sources may be one of the most promising means of accelerating tissue repair. In a recent study on the efficacy of flavonoid, which was efficiently induced in bio fabricated nano-biomaterials, flavonoid-loaded silver derived from the seed of Madhuca longifolia plant was found to be effective [168]. Compared to flavonoid-loaded gold and bimetallic, it enhanced wound healing by up to 80.33 %. In another recent study, chronic wounds treated with carbonized nanogel (copper sulphide nanoclusters) and quercetin exhibited fast healing of wounds. The discovered multifunctional nanogel can stimulate angiogenesis, epithelialization, and collagen synthesis to speed granulation tissue formation [169].”
|
|
8) |
Conclusion of the review: - Dear authors, I suggest a thorough revision of the organization of the manuscript. Several times it was possible to observe information about flavonoids in different paragraphs, making the text tiring, confusing, and not very explanatory. In this sense, I suggest that the text be organized so that there is an objective sequence of information. Ex: 1. Wounds and their morphological presentations 2. Conventional treatments 3. Socioeconomic effects of morbidity and mortality from chronic wounds 4. Inclusion of natural products as an alternative for treating wounds 5. Flavonoids 6. Mechanisms of action of flavonoids in wound repair 7. Antibacterial properties of flavonoids
|
Text organization has been thoroughly revised and organized according to the sequence as per the reviewer’s comment. We truly appreciate it.
Line 111 - Wounds and their morphological presentations Line 214 – Treatments for treating wound Line 319 - Socioeconomic effects of morbidity and mortality from chronic wounds Line 302 - Natural products as an alternative for treating wounds included Line 345 – Flavonoids Line 592 – Mechanisms of action or pathways of flavonoids in wound repair Line 565 - Antibacterial properties of flavonoids
|

Reviewer 3 Report
Healing of wounds is considered as a serious problem that affects the healthcare sector in many countries, primarily due to diabetes and obesity. It gets worse because of unhealthy lifestyles and habits.
Wound healing is a complicated physiological process that is essential for restoring the epithelial barrier after an injury.
Numerous studies have reported that flavonoids possess wound-healing properties due to their well-acclaimed anti-inflammatory, angiogenesis, re-epithelialization, and antioxidant effects.
They have been shown to be able to act on the wound healing process by expression of biomarkers respective to the pathways that mainly include Wnt/β-catenin, Hippo, TGFβ, Hedgehog, JNK, (Nrf2/ARE, NF-κB, MAPK/ERK, Ras/Raf/MEK/ERK, PI3K/Akt, NO and, etc...
Hence, the authors have compiled existing evidence on the manipulation of flavonoids towards achieving skin wound healing, together with current limitations and future perspectives in support of these polyphenolic compounds as safe wound healing agents in this review.
Author Response
REVIEWER 3:
Healing of wounds is considered as a serious problem that affects the healthcare sector in many countries, primarily due to diabetes and obesity. It gets worse because of unhealthy lifestyles and habits.
Wound healing is a complicated physiological process that is essential for restoring the epithelial barrier after an injury.
Numerous studies have reported that flavonoids possess wound-healing properties due to their well-acclaimed anti-inflammatory, angiogenesis, re-epithelialization, and antioxidant effects.
They have been shown to be able to act on the wound healing process by expression of biomarkers respective to the pathways that mainly include Wnt/β-catenin, Hippo, TGFβ, Hedgehog, JNK, (Nrf2/ARE, NF-κB, MAPK/ERK, Ras/Raf/MEK/ERK, PI3K/Akt, NO and, etc...
Hence, the authors have compiled existing evidence on the manipulation of flavonoids towards achieving skin wound healing, together with current limitations and future perspectives in support of these polyphenolic compounds as safe wound healing agents in this review.
Thank you very much for the constructive comment. We really appreciate it.

Reviewer 4 Report
The review paper submitted for review was prepared in a traditional style. I have a few comments:
- the authors repeat certain information too often in the manuscript, for example they state the phases of wound healing several times (one time is enough)
- flavonoids belonging to the broad class of polyphenols are the least stable compounds, please refer to the topic of flavonoids forming e.g. quinones,, is this form safe for cells?
- flavonoids have antioxidant activity, but can they also generate oxidative stress and how will this affect wound healing?
- too little attention is paid by the authors to the antibacterial properties of flavonoids, which are also an important element in wound healing
- please review the manuscript for editing, remove unnecessary spaces, the table signature should be on the same page as the table, etc.
Author Response
Manuscript ID: ijms-2143070
Manuscript Title: Flavonoids as Potential Wound Healing Molecules: Emphasis on Pathways
Perspective
REVIEWER 4:
The review paper submitted for review was prepared in a traditional style. I have a few comments:
Response To Reviewer:
We appreciate and wish to thank you for the useful reviewers’ comments, suggestions, and the opportunity to improve the manuscript. We have revised the manuscript, including those based on the comments/suggestions accordingly, and the detailed corrections are listed below in the authors’ response column. The changes in the manuscript are highlighted in yellow.
|
No. |
Reviewers’ comments |
Authors’ response |
|
1) |
The authors repeat certain information too often in the manuscript, for example they state the phases of wound healing several times (one time is enough) |
Line 66 – Corrected. Deleted phases of wound healing under subtopic 1: Introduction Line 112 - Maintain wound healing phases under subtopic 2: Wound Healing. |
|
2) |
Flavonoids belonging to the broad class of polyphenols are the least stable compounds, please refer to the topic of flavonoids forming e.g. quinones,, is this form safe for cells?
|
Line 370 - Reviewer’s comment has been duly addressed. “Flavonoids protect body cells from oxidative damage, which can cause disease, and they have advantageous defensive actions. The treatment of some quinone as flavonoids with a solvent containing water restored its potent antioxidant activity. For example, carnosol quinone is the antioxidation product of carnosol, which possesses a very weak antioxidant activity. Quinones are a class of toxicological intermediates that can cause a number of harmful consequences in vivo to cells, including acute immunotoxicity, cytotoxicity, and carcinogenesis. In contrast, quinones can produce cytoprotection by inducing detoxifying enzymes, anti-inflammatory actions, and altering redox state. The methods by which quinones exert these actions can be rather complicated [75].”
|
|
3) |
Flavonoids have antioxidant activity, but can they also generate oxidative stress and how will this affect wound healing?
|
Line 379 - Reviewer’s comment has been resolved.
“Many flavonoids have been characterised as potent reactive oxygen species (ROS) inhibitors, making them vital antioxidant food components. The influence of ROS on the oxidation of quercetin, kaempferol, morin, catechin, and naringenin was investigated. The reaction rates determined by spectrophotometry and oxygen consumption were drastically different. The quercetin possesses powerful antioxidant and anti-inflammatory effects, which support its potential use in wound healing. Additionally, quercetin can reduce both acute and chronic inflammatory stages. Reactive oxygen species and oxidative stress have a little role in the normal physiology of wound healing, but an excess of either can hinder healing. The use of antioxidants as most flavonoids is thought to speed up wound healing by reducing oxidative stress in the wound. Only some flavonoids with o-Quinones can generate oxidative stress [75].” |
|
4) |
Too little attention is paid by the authors to the antibacterial properties of flavonoids, which are also an important element in wound healing |
Line 565 – Corrected and more information have been added accordingly. Authors have paid more attention towards antibacterial properties of flavonoids in wound healing in subtopic 5.2. |
|
5) |
Please review the manuscript for editing, remove unnecessary spaces, the table signature should be on the same page as the table, etc.
|
Done and corrected throughout the manuscript. The manuscript has undergone a rigorous editing process, including the removal of extra spaces and the placement of the table signature on the same page as the table given. |

Round 2
Reviewer 2 Report
Dear authors, after reviewing the text, I consider the manuscript ready for editing and publication as suggested to reviewers.
Author Response
Dear Editors
Thank you for giving me the opportunity to submit a revised draft of my manuscript titled [Flavonoids as Potential Wound Healing Molecules: Emphasis on Pathways Perspective] to [IJMS]. We appreciate the time and effort that you and the reviewers have dedicated to providing valuable feedback on the manuscript. We are grateful to the reviewers for their insightful comments on the paper. We have been able to incorporate changes to reflect most of the suggestions provided by the reviewers. We have highlighted the changes within the manuscript.
Here is a point-by-point response to the reviewers’ comments and concerns.
Editor comments
The paper should do minor typos corrections:
- lines 78-83 different font color (it looks like it is grey instead of black
- line 201 remove extra space in the sentence
- line 255 complete Figure 2 caption (figure legend seems incomplete)
- lines 685-688 references 111 and 112 appear cut. Please correct.
- line 908 reference 79 appears with number repeated
The authors would like to thank you the reviewer for the valuable comments and recommendation
-
The revision has been done as requested. The correction is also done throughout the manuscript
Reviewer 2
Comments and Suggestions for Authors
Dear authors, after reviewing the text, I consider the manuscript ready for editing and publication as suggested to reviewers.
The authors would like to thank again the reviewer for the valuable comments and recommendation. We really appreciate your encouraging recommendation that push us to improve the review further
We look forward to hearing from you in due time regarding our submission and to respond to any further questions and comments you may have.
With very best regards,
Murni Nazirah Sarian, PhD
